# Epidemiological Characteristics of COVID-19 Cases in Non-Italian Nationals in Sicily: Identifying Vulnerable Groups in the Context of the COVID-19 Pandemic in Sicily, Italy

**DOI:** 10.3390/ijerph19095767

**Published:** 2022-05-09

**Authors:** Palmira Immordino, Dario Genovese, Fatima Morales, Alessandra Casuccio, Emanuele Amodio

**Affiliations:** 1Department of Health Promotion, Mother and Child Care, Internal Medicine and Medical Specialties, University of Palermo, Via del Vespro 133, 90127 Palermo, Italy; dario.genovese@unipa.it (D.G.); alessandra.casuccio@unipa.it (A.C.); emanuele.amodio@unipa.it (E.A.); 2Department of Preventive Medicine and Public Health, University of Seville, 41004 Sevilla, Spain; fmmarin@us.es

**Keywords:** access to health care, vulnerable populations, migrants, refugee and asylum seeker health care, inequalities in health and health care, health inequalities

## Abstract

As in other parts of the world, undocumented migrants in Italy suffer worse health status due to their immigration enforcement situation and other vulnerabilities such as precarious illegal jobs, exploitation and abuse or barriers to higher education, with higher prevalence of chronic noncommunicable diseases. The COVID-19 pandemic, as other pandemics, has not affected everyone equally. The undocumented was one of the most affected groups with regard to hospitalization rates and mortality worldwide. Sicily is one of the gates of entrance to Europe for migrants and asylum seekers from Africa and Asia. Herein, we described the epidemiological characteristics of COVID-19 cases in Sicily to compare hospitalization rate and mortality between Italian nationals and foreigners. We extracted data from the integrated national surveillance system established by the Italian National Institute of Health (Istituto Superiore di Sanità, ISS) to collect information on all COVID-19 cases and deaths in Sicily. We found that the hospitalization rates were higher in undocumented foreigners, and they were most likely to present a more severe clinical outcome compared to Italian nationals. Inclusive public health policies should take this population group into consideration to achieve the Health for All goal.

## 1. Introduction

COVID-19, declared a pandemic on 11 March 2020, has caused 281,808,270 confirmed cases, including 411,759 deaths, reported to WHO as of 29 December 2021 [1]. COVID-19 is reasonably one of the most devastating crises the world has faced since World War II. Refugees and migrants face similar health threats from COVID-19 as host populations [2]. Moreover, they may have higher vulnerabilities due to the conditions of their migratory journeys, poor nutritional status and limited access to safe food, water and sanitation, harsh living conditions and limited work opportunities due to their legal status [3].

In some countries, undocumented migrants may be excluded from national health programs and financial protection for health and social services [4]. On one hand, this exclusion might make it difficult for countries to reach this population with the COVID-19 response measures and strategies put in place, increasing the risk of outbreaks in these populations and posing an additional threat to public health [5]. On the other hand, migrants and refugees are exposed to consequences and the impact of immediate COVID-19 response measures, including worldwide lockdowns [6] that can be even more challenging for this population, with impacts on their jobs [7], standard of living, access to services including health care, food, education and women and girls’ safety in confined settings [8].

The pandemic caused a decrease in migration flows. Between January and November 2020, there were 114,300 illegal border crossings, 10% less than in the same period of 2019 [9]. However, variations between regions are significant. Indeed, data collected between March 2020 and February 2021 for the Central Mediterranean, which represents the most dangerous migration route, reported 34,100 arrivals (versus 11,500 in 2019) (+154%) with the majority arriving in Lampedusa [10].

Currently, almost 8.7 million foreign citizens are estimated to be living in Italy, 6.3 million of them represent Italian migrant stock [11].

The pandemic affected all aspects of migration, from migration flows to the quality of life of permanent migrants, making migrants more vulnerable and therefore at higher risk to COVID-19. Access to information might be limited due to language and cultural barriers. Since the beginning of the COVID-19 pandemic, stigma and discrimination have also increased with a negative influence on health behaviors as well as consequences on physical and mental health due to stigmatization [12].

Italy has been one of the most severely affected countries during the first wave of the pandemic. Since 15 January, 6,129,050 people have been diagnosed with the infection, and over 136,652 (2.2%) have died of it [13].

On one hand, considering the strong association between COVID-19-related deaths and older age and comorbidities [14], it is possible that newly arrived migrants may face a lower risk of being severely affected by COVID-19 because they are generally younger and healthier compared to the local population [15].

On the other hand, there is concern that underdiagnosis or delayed diagnosis in non-Italian populations may expose this population group to worse clinical outcomes compared with the local population [16].

From January to November 2021, a high increase in arrivals of refugees and migrants by sea was reported in Italy. A total of 62,943 people arrived in the first 11 months of 2021, almost two times the figures reported in 2020 (32,563) and four times the figure reported in 2019 (10,882). The vast majority of monthly sea arrivals (6100 persons) disembarked in Sicily, most commonly landing on the island of Lampedusa (3371 persons) [17].

This study aims to provide an overview of the epidemiological characteristics of COVID-19 cases among non-Italian nationals and compare them with Italian nationals in Sicily (Italy).

## 2. Materials and Methods

### 2.1. Data Sources

Data were extracted from the integrated national surveillance system established by the Italian National Institute of Health (Istituto Superiore di Sanità, ISS) to collect information on all COVID-19 cases and deaths occurring in Sicily.

The study analyzes mortality rates and clinical characteristics of Italian and foreign COVID-19 patients between February 2020 and April 2021. Specifically, 162,260 medical records were collected. The status of Italian native or foreign was assigned based on the individual’s country of birth, indicated in the relevant field of the medical record, or extrapolated from the fiscal code.

Data, in fact, as in other countries, are not disaggregated by migratory status.

The following data were extracted: demographics; comorbidities; symptoms at onset; hospitalization; treatments received, including intubation; admission to intensive care unit (ICU); time from onset to hospitalization, death, and time from hospitalization to death.

To define the clinical outcome, we established the following severity index:Non-severe clinical outcome (NSCO);Severe clinical outcome (SCO).

In particular, severe clinical outcome was defined as the occurrence of at least one of the following events: hospitalization, intubation, admission to ICU, death. Non-severe clinical outcome included the remaining conditions (asymptomatic or mild conditions).

World countries—and, subsequently, relative citizens of these countries—were categorized according to the ISTAT classification in:Highly developed countries (European countries except those of Central–Eastern Europe, North America, Israel, Japan and Oceania);European countries with High Migratory Pressure (Central and Eastern Europe);Non-European countries with High Migratory Pressure (Africa, Asia, Central and South America).

Italy was not included in any category: it was considered as a separate group.

### 2.2. Statistical Analysis

Normality distribution of quantitative variables was assessed by the adjusted Jarque–Bera test (N > 5000) and, accordingly, all variables that were normally distributed have been presented as mean (SD), whereas non-normally distributed variables have been summarized as median (IQR).

Categorical variables have been summarized as absolute number (percentage).

Differences in the distribution of proportions by group were analyzed using the chi-square test, while non-parametric continuous variables were compared by using the Mann–Whitney U test.

A multivariate logistic regression analysis was used to calculate the odds ratio (OR) and 95% confidence interval (CI) with respect to the factors that may determine the worsening of the clinical scenario in the context of SARS-CoV-2 infection. For this reason, the abovementioned model evaluated the contribution of different independent variables (sex, age, ISTAT classes, onset trimester, days between first day of clinical symptoms and first swab) in determining the risk of developing a Severe Clinical Outcome (using as a reference level the Non-severe Clinical Outcome).

Statistical significance was defined as *p* ≤ 0.05. Analyses were performed using R Software analysis, version 4.0.5, R Foundation for Statistical Computing (Vienna, Austria).

## 3. Results

After exclusion of incomplete records or patients where relevant information was not available, a total of 162,260 cases were included in the analysis. Of these, 155,441 were Italian nationals (95.8%), 531 (0.3%) were undocumented non-Italians, and 6288 (3.8%) were foreigners with a tax identification number indicating that 2964 (1.8%) were from non-European countries with High Migratory Pressure (nE-HMP), 1159 (0.7%) from European countries with High Migratory Pressure (E-HMP) and 2165 (1.3%) from Highly Developed Countries (HDCs).

### 3.1. Socio-Demographic Characteristics and Time of Diagnosis

Among all COVID-19 cases, compared with Italian nationals (75,687; 48.7%) undocumented non-Italians and foreigners from nE-HMP were more frequently males (391; 73.6% and 1859; 62.7%, respectively). Foreigners from E-HMP and HDCs, were more frequently females (810; 69.9% and 1261; 58.2%, respectively). Apart from cases from HDCs, where the median age was 47.7 years (39.2–45.7), all the other cases among non-Italian nationals were younger than Italian cases, whose median age was 46.4 years (27.8–45.6), especially those undocumented (median age: 25.4 years (18.1–29.8). In terms of clinical outcome, we identified four possible outcomes (i.e., Non-hospitalized, Hospitalized, Intubation, Death) that were grouped as “Worst Clinical Outcome” (see Table 1). Each individual was then assigned to her/his worst outcome. According to this classification, death was the worst outcome in 3% of the Italian cases, while 1.7% of the undocumented non-Italians, 1.7% of the nE-HMP, 0.6% of the e-HMP and 0.6% of the cases from HDCs died from COVID-19. However, we found that the hospitalization rate was significantly higher in cases among undocumented non-Italian nationals (13.4%).

A complete description of demographic and clinical characteristics is shown in Table 1.

### 3.2. Univariable Analyses

Males were more likely to have SCO than females (11.2% vs. 8%, *p* < 0.001). Regarding the categories classified by ISTAT, undocumented non-Italians were more likely to have a severe clinical outcome than Italians (15.5% vs. 9.6%, *p* < 0.001). The median age was found to be higher in subjects with SCO (70.8 vs. 43.5 years, *p* < 0.001). The number of positive cases with severe clinical outcomes decreased over time (42% in January–March 2020 vs. 8% in January–March 2021). The abovementioned results are summarized in Table 2.

In Figure 1 the differences in the median number of days between first swab, symptoms onset and hospitalization among Italians and non-Italians with COVID-19 are represented. Among Italians, HDCs and E-HMPs, nasal swabs were in median performed two days before symptom onset. In contrast, nE-HMPs and undocumented non-Italians received a positive result right before symptom onset or even the same day as the clinical outcomes occurred. For those who were hospitalized, it is interesting to notice that undocumented non-Italians were hospitalized sooner than the other groups.

### 3.3. Multivariable Analysis

According to the multivariable logistic regression analysis, the odds of SCO were higher in older subjects—as each unitary increment per age leads to an adj-OR of 1.07 (95% CI = 1.069–1.072—male patients (ref. females; adj-OR = 1.96, 95% CI = 1.87–2.059), and in patients with a larger interval between the first day of clinical symptoms and the first swab (increment per unit; adj-OR = 1.096, 95% CI = 1.09–1.10). With regard to ISTAT classification groups, taking as referent Italians, it was shown that the odds were statistically significantly higher in undocumented non-Italians (OR = 15.73; *p* < 0.001) and those who were born and raised in nE-HMP (OR = 2.63; *p* < 0.001) and in E-HMP (OR = 1.56; *p* < 0.001). A significant reduction in the odds was found in those who came from HDCs (OR = 0.68; *p* < 0.001). Similarly, the odds of SCO were progressively reduced over time when compared to the January–March 2021 (*p* < 0.001) (Figure 2).

## 4. Discussion

The COVID-19 pandemic had a primary role in emphasizing health disparities among certain vulnerable groups; these disparities are probably the result of long-standing structural inequities.

According to international studies, the individuals with the most severe symptoms and the highest risk of death are the elderly and those with chronic illness [18]. This study confirms that older age, male sex and a diagnosis made in the early pandemic trimesters are risk factors for a severe clinical outcome.

As for the trimesters of SARS-CoV-2 infection diagnosis, the results suggest that poor health system preparedness towards public health emergencies led to excess morbidity and mortality for COVID-19. In this sense, that increased risk of SCO in the first trimester—a risk that progressively decreased over the trimesters—could be attributable to both higher mortality and lesser restricting criteria for hospitalizations among symptomatic COVID-19 patients. The higher hospitalization rate has to be considered as an act of greater precaution against a disease whose natural history was not very well known.

Although the incidence of non-hospitalized cases with clinical outcomes is similar between foreigners and the general resident population (84–94%), as shown in another study carried out in Italy [19], the percentage of hospitalized cases was higher in undocumented (13.4%) and nE-HMP foreigners (9.5%) than in our study (4.5–6.4%) and in the one carried out by Fabiani et al. [16].

Another relevant result concerns the risk of a severe clinical outcome in undocumented non-Italian patients more than 15 times higher than in the Italians included in this study. International literature widely confirms this result and some of the potentially leading factors have been reported in the introduction previously discussed. According to Hintermeier at al [20], the incidence of SARS-CoV-2 infection has been shown to be higher among migrants, refugees, asylum seekers and IDPs. With regards to hospitalization, there is not yet a comprehensive overview of the extent to which foreign-born living temporarily or permanently in different HICs show hospitalization rates.

For example, in Denmark, where non-Western migrants represent less than 10% of the total population, they accounted for 15% of total COVID-19 hospital admissions [21].

On the other hand, in Greece, one of the countries hosting most of the migrants and refugees in the Mediterranean region, almost 50% of COVID-19 hospitalized patients were refugees and migrants from the capital city [22]. According to the Italian surveillance system, non-Italian cases had a 39% increased risk of being hospitalized and a 19% increased risk of being admitted to an intensive care unit, especially in those coming from nE-HMP [19].

The higher observed incidence and severity of COVID-19 in ethnic minority groups and some migrants could probably be due to the complex interaction of socio-economic health determinants, barriers to accessing care and higher prevalence of underlying medical comorbidities that lead to more severe disease [23]. However, also regarding severity and excess death for COVID-19, there is not a clear picture across the countries where evidence is available.

A study from Canevelli et al. [24] shows that proportions of natives and migrants among the COVID-19-related deaths (97.5% and 2.5%, respectively) were similar to the relative all-cause mortality rates estimated in Italy in 2018 (97.4% and 2.6%, respectively).

In Sweden, an analysis of all recorded COVID-19 deaths found that being a migrant from a nE/HMP is predictive of a higher risk of death from COVID-19 [25]. However, in Denmark, no differences were observed in mortality from COVID-19 by migration status [21].

Our study confirms what was found in Italy, where, according to the Italian surveillance system, a 32% increased risk of death was found in non-Italians from nE/HMP [16]. The clinical phenotype of non-Italian nationals dying with COVID-19 was similar to that of natives except for the younger age at death. International migrants living in Italy do not have a mortality advantage for COVID-19 and are exposed to the same risk of poor outcomes as their native counterparts.

The last aspect we explored is the time elapsing between the first swab, the symptom onset and a possible hospitalization. The period between the first swab test and first day of clinical presentation is equal to 0 days among undocumented non-Italians, while it is two days among Italians. This data suggest that it is easier for Italians and, in general, for residents to access the COVID-19 surveillance systems and, therefore, to perform the first swab with a median advance of two days. One study observed how the Latino community in the U.S. suffer barriers to diagnostic testing, such as fear of job/income loss, discrimination/stigmatization, cost, and accuracy of testing [26]. Identifying the infection early allows for the prevention of the onset of complications from SARS-CoV-2 infection [27]. Finally, looking at the difference in the median number of days between first swab, symptom onset and hospitalization among Italians and non-Italians with COVID-19, it is worth noting that the period between the first swab and the hospital admission is 13 days among Italians, while it is 11 days in undocumented foreigners. It is possible to hypothesize that this lower interval may be attributable to a delay in the detection of the viral infection and, consequently, to a condition that leads to hospitalization in a shorter time.

In order to confirm these hypotheses, in the study by Fabiani et al. exploring the first wave of the pandemic (February to July 2020) [16], 7.5% cases were identified in non-Italian nationals, and they were diagnosed at a later stage of the infection. Therefore, they were more likely to be hospitalized and admitted to the ICU, with differences being more pronounced in those coming from countries with a lower human development index (HDI). An increased risk of death in non-Italian cases from low-HDI countries was also observed.

However, the results need to be contextualized as they may have some limitations.

First, in this study it was not possible to stratify the sample for the socio-economic deprivation index or the presence of comorbidities and lifestyle risk factors; all the considerations made could have a slightly different weight.

Second, it was not possible to evaluate in detail the severity of symptoms among the different groups; this limitation is due to the lack of data as well as data-entry misclassification. For this reason, we decided not to consider this parameter, and we only used the clinical outcome, free from this type of errors.

## 5. Conclusions

Results of our study confirm the role that different risk factors play towards severe clinical outcomes, including older age, male sex and a diagnosis in the first trimesters of the COVID-19 pandemic. Furthermore, it has been confirmed that compared to Italians undocumented foreigners have a greater risk of severe clinical outcomes. It remains to be understood whether the increased hospitalization rate in these patients may be linked to a difficulty in accessing diagnosis in the pre-clinical phase and to what extent inequalities and socio-economic deprivation have contributed to determining the risk.

According to IOM [28], it is mandatory to refine domestic legal and policy frameworks to improve early detection and treatment in migrants, because excluding them may increase the transmission risk and may negatively impact the outcome. This inclusive approach at a global level can only be achieved if the mechanisms of health care are clear at all levels, and authorities are able to understand and respond to the needs of migrants and refugees. Recovering from the COVID-19 pandemic must be seen as an opportunity to reduce inequities and develop inclusive policies for accessing health systems by vulnerable group populations [29].

## Figures and Tables

**Figure 1 ijerph-19-05767-f001:**
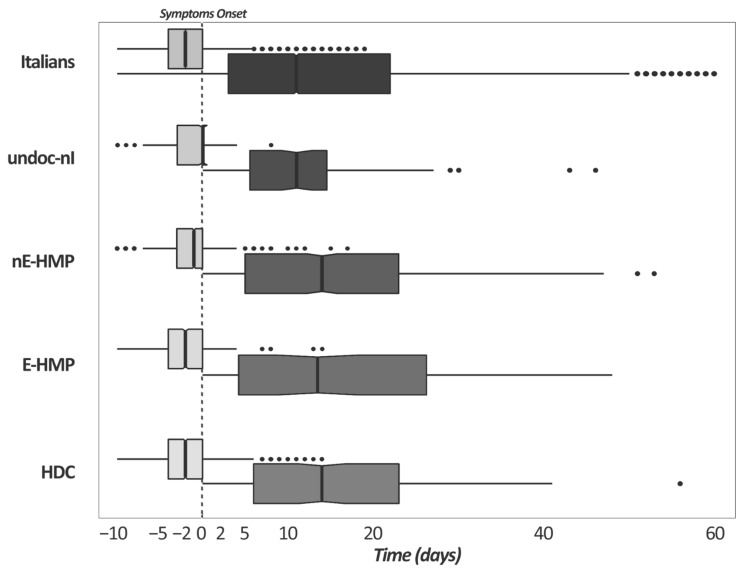
Difference in the median number of days between first swab, symptom onset (light gray box plots), and between symptom onset and hospitalization (dark gray box plots) among Italians and non-Italians with COVID-19. Dotted lines represent the symptom onset (time 0 days). **Legend:** Italians; undoc-nI = undocumented non-Italians; nE-HMP = non-Italians from non-European countries with High Migratory Pressure; E-HMP = non-Italians from European countries with High Migratory Pressure; HDC = non-Italians from Highly Developed Countries.

**Figure 2 ijerph-19-05767-f002:**
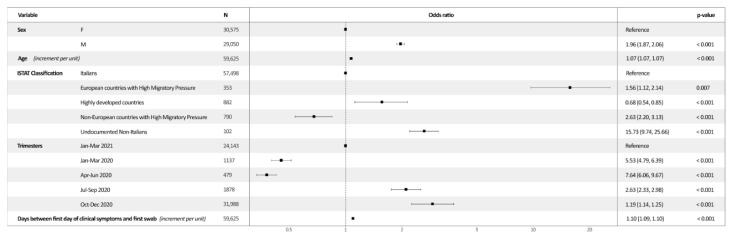
Visual representation of multivariable logistic regression and odds ratio.

**Table 1 ijerph-19-05767-t001:** Socio-demographic and clinical characteristics of the COVID-19 cases included in the study.

	Italians N = 155,441 (95.8%)	Undocumented Non-Italians N = 531 (0.3%)	Non-Italians from N = 6288 (3.9%)		
			**nE-HMP** 2964 (1.8%)	**E-HMP** 1159 (0.7%)	**HDCs** 2165 (1.3%)
**Sex, N (%)**					
**- F**	79,754 (51.3%)	140 (26.4%)	1105 (37.3%)	810 (69.9%)	1261 (58.2%)
**- M**	75,687 (48.7%)	391 (73.6%)	1859 (62.7%)	349 (30.1%)	904 (41.8%)
**Age, Median (IQR)**	46.4 (27.8–45.6)	25.4 (18.1–29.8)	36.9 (25.2–39.3)	38.9 (28.9–39.3)	47.7 (39.2–45.7)
**Stratification per Trimester, N (%) ^1^**					
**- Jan–Mar 2020**	1717 (1.1%)	1 (0.2%)	6 (0.5%)	13 (0.4%)	17 (0.8%)
**- Apr–Jun 2020**	1223 (0.8%)	9 (1.7%)	4 (0.3%)	25 (0.8%)	13 (0.6%)
**- Jul–Sep 2020**	3502 (2.3%)	182 (34.3%)	65 (5.6%)	339 (11.5%)	74 (3.4%)
**- Oct–Dec 2020**	86,645 (55.9%)	162 (30.6%)	480 (41.6%)	1342 (45.4%)	1256 (58.1%)
**- Jan–Mar 2021**	61,874 (39.9%)	176 (33.2%)	600 (51.9%)	1240 (41.9%)	800 (37.0%)
**Worst Clinical Outcome, N (%) ^1^**					
**- Non-hospitalized**	140,199 (90.4%)	446 (84.5%)	2621 (88.8%)	1090 (94.2%)	2044 (94.6%)
**- Hospitalized**	9987 (6.4%)	71 (13.4%)	279 (9.5%)	60 (5.2%)	98 (4.5%)
**- Intubation**	113 (0.1%)	2 (0.4%)	1 (0%)	0 (0%)	2 (0.1%)
**- Death**	4820 (3.1%)	9 (1.7%)	50 (1.7%)	7 (0.6%)	16 (0.7%)

^1^ Some data are missing; nE-HMP = non-European countries with High Migratory Pressure; E-HMP = European countries with High Migratory Pressure; HDCs = Highly developed countries.

**Table 2 ijerph-19-05767-t002:** Socio-demographic characteristics and risk factors associated with severe clinical outcome.

	Non-Severe Clinical Outcome	Severe Clinical Outcome	*p*-Value
**Sex, N (%)**			
**- F**	76,258 (92.0%)	6643 (8.0%)	<0.001
**- M**	70,142 (88.8%)	8872 (11.2%)	
**Age, Median (IQR)**	43.51 (26.1–58.2)	70.77 (57.3–81.6)	<0.001
**ISTAT classification, N (%)**			
**- Italians**	140,199 (90.4%)	14,920 (9.6%)	<0.001
**- Undocumented Non-Italians**	446 (84.5%)	82 (15.5%)	
**- Non-Italians from non-European countries with High Migratory Pressure**	2621 (88.8%)	330 (11.2%)	
**- Non-Italians from European countries with High Migratory Pressure**	1090 (94.2%)	67 (5.8%)	
**- Non-Italians from highly developed countries**	2044 (94.6%)	116 (5.4%)	
**Stratification per Trimester, N (%)**			
**- Jan** **–Mar 2020**	1018 (58.0%)	736 (42.0%)	<0.001
**- Apr** **–Jun 2020**	866 (68.0%)	408 (32.0%)	
**- Jul** **–Sep 2020**	3487 (83.8%)	675 (16.2%)	
**- Oct** **–Dec 2020**	81,579 (90.8%)	8271 (9.2%)	
**- Jan** **–Mar 2021**	59,260 (92.0%)	5140 (8.0%)	

## Data Availability

Not applicable.

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
