# Peer review of "Epidemiological Characteristics of COVID-19 Cases in Non-Italian Nationals in Sicily: Identifying Vulnerable Groups in the Context of the COVID-19 Pandemic in Sicily, Italy"

_ijerph, 2022, doi:10.3390/ijerph19095767_

Round 1
Reviewer 1 Report
In this paper the authors present a statistical analysis of the incidence of Covid19 virus on stratified data from Ministero della Salute in Italy. Specifically, they tried to find any substancial difference in the number of affected individuals between the migrants from Africa and Asia compared and the italian resident population. This works is sound and is expected to be of interest for a broad audience. The statistical analysis is well conducted with the usage of ad-hoc tests and the obtained results effectively show the declared conclusions. Nonetheless, there are a few issues that should be addressed by the authors prior for the manuscript to be accepted in the journal.
Line 63. Use “On one end ” instead of
Line 79. Please provide a link to the dataset used. The data used should be reachable and used for any comparison/control.
Line 70-73: this text was copied from “https://reliefweb.int/report/italy/italy-sea-arrivals-dashboard-november-2021#:~:text=In%20the%20first%20eleven%20months,in%202020%20and%202019%2C%20respectively.”. Please consider to resentence it.
Line 122. Please make available the R code used for these analyses through platforms such as GitHub or any other similar.
Author Response
Many thanks for your valuable comments. Please see the attachment.

Reviewer 2 Report
The downside of the study is the division of patients based on the country of birth only.
"Statistical analysis" section: To compare two continuous variables, the Mann-Whitney test should be used instead of the Kruskal-Wallis test. The model type is incorrectly selected. Use a logistic regression model instead of multinomial logistic regression.
Table 1. Explain: "worst clinical outcame"
Figure 1: This is not a correct representation of the results that compare the two groups. I suggest to use a “box plot” charts. There is no description of Fig.1. Describe these results in section 3.2 (these are one-dimensional results).
Figure 2. Is this a multivariate model? This information should be added. Expand the description of these results. Remember to indicate the reference category. Correctly describe the continuous variables ("increase per unit").
A logistic regression model estimates OR, that is, use the term "odds" should be instead of "risk".
Author Response

(The authors gave the same response as above.)

Round 2
Reviewer 2 Report
Figure 1 - Please definitely make box plots for both groups or a different type of chart, but should present both groups separately. The chart can show the median and quartiles (without minimum and no maximum).
Using "odds" - I agree with you, if the risk (<0.1) is low you can use OR as an approximation to RR. However, in this study, the risk of NSCO is likely> 0.1 for COVID sufferers. Odds should be used in this situation. If R <0.1, show it.
Author Response
Many thanks once again for your time and suggestions. Please see the attachment.
